# CAMP: an instrumented platform for balloon-borne aerosol particle studies in the lower atmosphere

Christian Pilz[1], Sebastian Düsing[1], Birgit Wehner[1], Thomas Müller[1], Holger Siebert[1], Jens Voigtländer[1], and Michael Lonardi[2]

[1]Leibniz Institute for Tropospheric Research, Leipzig, 04318, Germany
[2]Leipzig Institute for Meteorology, University of Leipzig, 04103, Germany

*Correspondence to*: Christian Pilz (pilz@tropos.de)

**Abstract**

Airborne observations of vertical aerosol particle distributions are crucial for detailed process studies and model improvements. Tethered balloon systems represent a less expensive alternative to aircraft to probe shallow atmospheric boundary layers (ABL). This study presents the newly developed cubic aerosol measurement platform (CAMP) for balloon-borne observations of aerosol particle microphysical properties. With an edge length of 30 cm and a weight of 9 kg, the cube is an environmentally robust instrument platform intended for measurements at low temperatures, with a particular focus on applications in cloudy Arctic ABLs. The aerosol instrumentation onboard CAMP comprises two condensation particle counters with different lower detection limits, one optical particle size spectrometer, and a miniaturized absorption photometer. Comprehensive calibrations and characterizations of the instruments were performed in laboratory experiments. The first field study with a tethered balloon system took place at the Leibniz Institute for tropospheric research (TROPOS) station in Melpitz, Germany, in the winter of 2019. At ambient temperatures between -8 and 15°C, the platform was operated up to 1.5 km height on 14 flights under a clear sky and cloudy conditions. The continuous aerosol observations at the ground station served as a reference for evaluating the CAMP measurements. Exemplary profiles are discussed to elucidate the performance of the system and possible process studies. Based on the laboratory instrument characterizations and the observations during the field campaign, CAMP demonstrated the capability to provide comprehensive aerosol particle measurements in cold and cloudy ABLs.

## 1 Introduction

The importance of atmospheric aerosol particles to the earth's climate, given by their direct and indirect effect on the earth's radiative budget, is widely known (Bond et al., 2013; Penner et al., 2004). Airborne aerosol observations provide valuable in situ information on particle properties and spatial distributions required for improvements in process understanding and model advancement, in particular for remote areas like the Arctic (Abbatt et al., 2018; Samset et al., 2014; Schacht et al., 2019; Schmale et al., 2021; Willis et al., 2018). The extensive NETCARE (Network on Climate and Aerosols: Addressing Key Uncertainties in Remote Canadian Environments) aircraft campaign above the Canadian Arctic, for instance, enabled the identification of vertically varying particle properties and source regions inside and outside the Arctic ABL (Willis et al., 2019). Furthermore, substantial vertical variability of long-range transported black carbon (BC) was observed, especially at higher altitudes (Schulz et al., 2019). Detailed process studies of new particle formation (NPF) at the cloud top with subsequent particle growth (Burkart et al., 2017; Leaitch et al., 2016; Willis et al., 2017) also rest upon the NETCARE aircraft observations. Still, the open question remains whether long-range transport or local sources are more relevant for particle abundance in the Arctic. More vertical particle distribution observations in the shallow and cloudy ABL are urgently needed to address this topic.

The limited observational capabilities of aircraft in the Arctic ABL due to frequently occurring low-level mixed-phase clouds highlight the need for a different approach to complement aircraft observations. Tethered balloon systems (TBS) demonstrated particular capabilities for inside cloud operations under light icing conditions in the Arctic (Creamean et al., 2020; Dexheimer et al., 2019; Egerer et al., 2019; Ferrero et al., 2016, 2019; Mazzola et al., 2016; Moroni et al., 2015). The high vertical

resolution of TBS allows for detailed vertical aerosol distribution measurements, thus, bridging ground with aircraft observations. Moreover, the temporal evolution of aerosol layers can be observed with TBS due to their ability to hover at a constant altitude (Jensen et al., 2002). A disadvantage of TBS and uncrewed airborne systems, in general, is the restricted payload that limits to lightweight instruments or custom-built devices (Bates et al., 2013; Boer et al., 2018; Creamean et al., 2018; Porter et al., 2020; Telg et al., 2017; Zinke et al., 2021).

With the weight restrictions given by the TBS, the focus is on composing a proper configuration of mobile devices to cover the most relevant microphysical properties of Arctic aerosol particles at high accuracy. Observations of nucleation mode particles originating from NPF in a weight-reduced configuration require a setup of two condensation particle counters (CPCs) with different lower detection limits (Heintzenberg et al., 1999; Hermann and Wiedensohler, 2001). Although many portable particle counters have been developed in recent years (e.g., Testo DiSCmini, Naneos Partector, Oxility NanoTracer), there is still a lack of commercially available lightweight CPCs with low uncertainties that are internally recording time series of particle number concentration ($N$). Measurements of the Arctic aerosol particle number size distribution (PNSD) that is dominated by Aitken and Accumulation mode particles (Tunved et al., 2013) demand optical particle size spectrometers (OPSS) with low detection limits. The low BC concentrations in the Arctic ABL that are often slightly above the detection limit of conventional full-size instruments (Backman et al., 2017) are a particular challenge for mobile devices. In the context of airborne BC observations, the required low detection limits at long averaging intervals contradict desired high spatial coverage at short intervals (Pikridas et al., 2019). Finally, customized protective housing and a heating system are obligatory for operating sensible instruments inside clouds and at low ambient temperatures.

This study addresses the need for more vertical aerosol observations in the Arctic ABL by developing an instrumented platform for balloon-borne applications. The presented cubic aerosol measurement platform (CAMP) encases four mobile devices in a temperature-controlled and environmentally robust housing. Detailed calibrations and characterizations of the instruments were performed in laboratory studies at the World Calibration Center for Aerosol Physics (WCCAP) to ensure traceability and quality assured measurements. CAMP was tested and evaluated in a first field campaign with the BELUGA (Balloon-bornE modular Utility for profilinG the lower Atmosphere, Egerer et al., 2019) TBS at the Leibniz Institute for tropospheric research (TROPOS) station in Melpitz, Germany, in the winter of 2019. A case study from the campaign highlights the observational capabilities of CAMP and establishes a relation between lower tropospheric particle layers, ABL dynamics, and a sudden increase of nucleation mode particles on the ground.

## 2 CAMP system

### 2.1 Technical Design

CAMP is a lightweight and environmentally robust instrument payload designed for in situ measurements of aerosol particle microphysics with TBS under cold weather conditions (Figure 1). With a dimension of 35 x 35 x 35 cm and a total weight of 9 kg, the platform contains two CPCs, one OPSS, and an absorption photometer (Table 1). CAMP's frame structure is made of anodized aerospace aluminum. Carbon fiber composite sandwich plates of 3 mm thickness serve as the outer shell. Fasteners (Benloc Fastener Technik GmbH & Co. KG, Germany) hold the front and back panel for access to the instruments inside. Two fixations for four 3 mm stainless steel cables are integrated into the four vertical edges of the cube. The cables are connected to carabiners above and below CAMP to hook them into slings in the balloon tether.

Stable measurement conditions inside the cube are maintained at temperatures above 20°C by insulation and a controlled heating system. A 19 mm Armaflex (Armacell GmbH, Germany) insulation material was chosen because of its very low heat conductivity compared to other materials. In addition, Armaflex's high resistance to water vapor diffusion inhibits condensing water inside of CAMP, which is a potential risk during operations inside clouds. The controlled heating system (TR12-G, Telemeter Electronic GmbH, Germany) consists of a heating film on a custom aluminum plate, a blower, and a controller with a PT-100 temperature sensor.

The aerosol sampling system upstream of the instruments consists of a vertical funnel inlet, a silica-based diffusion dryer, and a flow splitter with an integrated core sampling system. The two CPCs and the OPSS run on a shared vacuum scroll pump

(model V05H012A, Air Squared Manufacturing Inc., USA) combined with customized critical orifices while the STAP runs on an internal pump. The critical pressure drop across the orifices is constantly monitored to assure constant flow rates under changing ambient pressures during balloon flights. Sample air temperature ($T$) and relative humidity (RH) are monitored with two sensors (HYT-939, B+B Thermo-Technik GmbH, Germany); one is located outside the platform downstream of the inlet and the other one inside CAMP downstream of the dryer. Another sensor (BME 280, Bosch Sensortec GmbH, Germany)

acquires barometric pressure ($p_b$), $T$, and RH. Time and position data is provided by a satellite receiver (Navilock NL-8004U, Tragant Handels- und Beteiligungs GmbH, Germany). A microcontroller (Teensy 3.6, PJRC.COM, LLC., USA) records the CPC and sensor data at 1 Hz on an inbuilt micro SD Card. The CPC data along with the barometric pressure is also transmitted via a second microcontroller (Seeeduino V4.2, Seeed Technology Co., Ltd., China) with a radio module on 868 MHz (Grove Long Range, Seeed Technology Co., Ltd., China) for real-time display on the ground during balloon flights. CAMP can

independently operate a minimum of three hours on one shared battery, depending on ambient temperatures and resulting needs for heating. Two identical CAMPs with interchangeable instrument slots were developed. A detailed technical sketch of the complete system with specifications for sampling lines, materials, scales, and electronics is provided in Figure S1 in the supplementary.

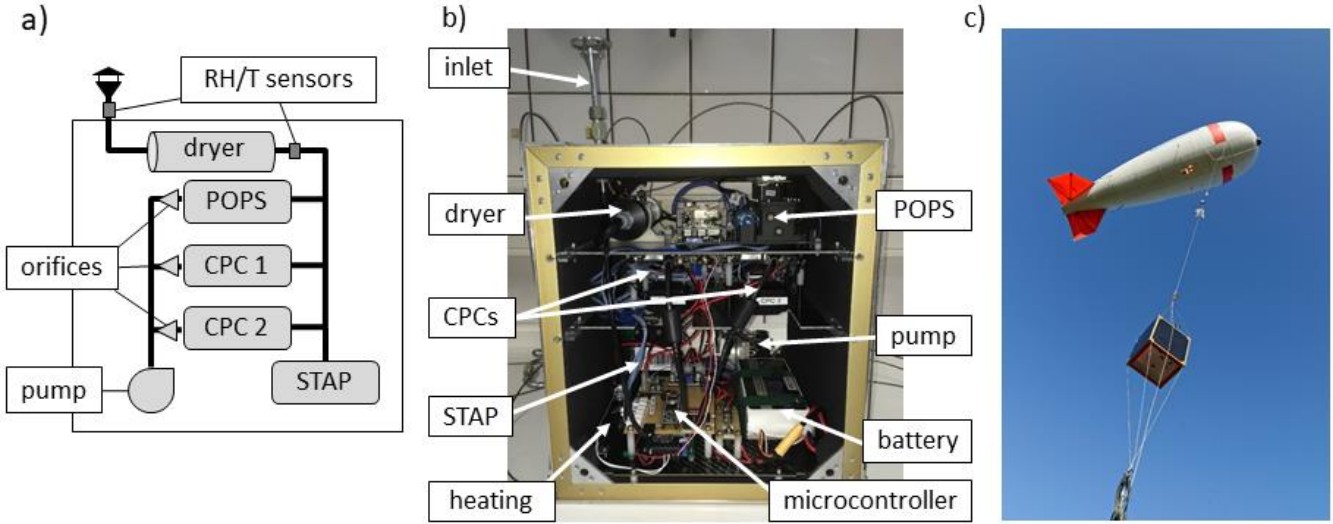

**Figure 1** a) Schematic sketch of the aerosol sampling system and instruments, b) overview of CAMP with the main components, and c) CAMP operated with BELUGA during the test campaign at Melpitz in February 2019. A detailed technical sketch of CAMP is provided in Figure S1 in the supplementary.

**Table 1** Characteristics of the aerosol instrumentation inside CAMP

| Instrument | Measured quantity | Range | Uncertainty | Sampling rate | Flow rate |
|---|---|---|---|---|---|
| Portable Optical Particle Spectrometer (POPS, Handix) | Particle number size distribution $N_{>150}$, $D_P = 0.15$ to 3 µm | 0 to 1200 cm$^{-3}$ | $N_{>150}$: ± 10 % $D_P$: ± 10 % | 1 s | sample: 0.2 l min$^{-1}$ sheath: 0.4 l min$^{-1}$ |
| Condensation particle counter (modified TSI 3007) | Particle number concentration $N_9$, $D_P = 0.009$ to 2 µm | 0 to $10^5$ cm$^{-3}$ | $N_9$: ± 10 % | 1 s | 0.11 l min$^{-1}$ |
| Condensation particle counter (modified TSI 3007) | Particle number concentration $N_{12}$, $D_P = 0.012$ to 2 µm | 0 to $10^5$ cm$^{-3}$ | $N_{12}$: ± 10 % | 1 s | 0.11 l min$^{-1}$ |
| Single Channel Tricolor Absorption Photometer (STAP model 9406, Brechtel) | Particle light absorption coefficient $\sigma_{abs}$(450, 525, 624 nm) | > 0.2 Mm$^{-1}$ | $\sigma_{abs}$: ± 10 % | 1 to 120 s | 1 l min$^{-1}$ |

## 2.2    Condensation Particle Counters

A well-characterized handheld CPC (model 3007, TSI Inc., USA) (Asbach et al., 2012; Hämeri et al., 2002; Mordas et al., 2008) was adapted for the instrumentation of CAMP, similar to Altstädter et al. (2015). The weight of the CPCs was reduced from 1.5 to 0.7 kg by removing the housing, display, and batteries. An external data acquisition was established with a microcontroller over the serial interface for concentration recordings at 1 Hz. The initial flow system was substituted with an external vacuum scroll pump and a customized critical orifice to set a constant sample flow rate. The slightly different flow

rate across the orifice introduced a constant offset in instrument counting efficiency that was considered with a correction factor determined by calibrations (details below). With the improvements in the flow system, the measurement uncertainties were reduced from ±20 to ±5 % for 10 s average particle number concentration relative to a calibrated reference Electrometer (3068B, TSI Inc., USA) for 40 nm silver particles.

Two CPCs with different lower cut-offs are commonly used on airborne platforms to detect nucleation mode particles

originating from NPF by calculating the difference in concentrations between the instruments. The lower detection limit is defined as the particle diameter ($D_{P50}$) at which the CPC shows 50 % counting efficiency. The $D_{P50}$ depends on the temperature difference ($\Delta T$) between the instrument's saturator and condenser (Banse et al., 2001). For the CPC 3007 on CAMP, varying $\Delta T$ were investigated with software commands to prevent hardware modifications from affecting device characteristics. Other than for full-size laboratory CPCs (Banse et al., 2001), model 3007 does not allow for individual temperature settings for the

saturator and condenser because of its combined warming/cooling system with Peltier elements. Therefore, differing $\Delta T$ can only be indirectly achieved with variations of the supply voltage of the Peltier elements yet without any absolute temperature information provided by the instrument. With external temperature sensors (TSIC 506, B+B Thermo-Technik GmbH, Germany) attached to the saturator and condenser, the standard and maximum voltage settings were measured to result in a $\Delta T$ of 11 K and 15.1 K, respectively. The corresponding $D_{P50}$ at the maximum $\Delta T$ of 15.1 K was determined in calibration with

silver particles at the World Calibration Center for Aerosol Physics (WCCAP) in a setup according to Wiedensohler et al. (2018). An implemented sintering process generates almost spherically shaped silver particles (Tuch et al., 2016). All tubing lengths are identical to avoid different diffusional losses. The counting efficiencies were measured against a reference Electrometer (3068B, TSI Inc., USA) over a five-minute mean at particle number concentrations of 1000 cm$^{-3}$. The particle mobility diameters ($D_P$) were size selected with a differential mobility analyzer (DMA) in a range from 5 to 40 nm. A $D_{P50}$ of

8 nm was found with a non-linear regression following Eq. (1):

$$Eff = Eff_{max}\left[1 - exp\left(\frac{D_{P0} - D_P}{D_{P50} - D_{P0}} \cdot ln2\right)\right] \tag{1}$$

with the maximum counting efficiency ($Eff_{max}$) and the particle diameter for zero counting efficiency ($D_{P0}$). Hämeri et al. (2002) found a $D_{P50}$ of 10 nm for the CPC 3007 at standard settings. A difference in detection limits of one CPC at standard and one at maximum settings would be relatively small with 8 and 10 nm. Therefore, an increase in $D_{P50}$ with lower Peltier

voltage settings resulting in a lower $\Delta T$ of 9.5 K was evaluated. The results of another calibration with four modified CPCs at $\Delta T$ of 15.1 K, 11 K, and 9.5 K in the WCCAP are displayed in Figure 2, with the fitting parameters given in Table 2.

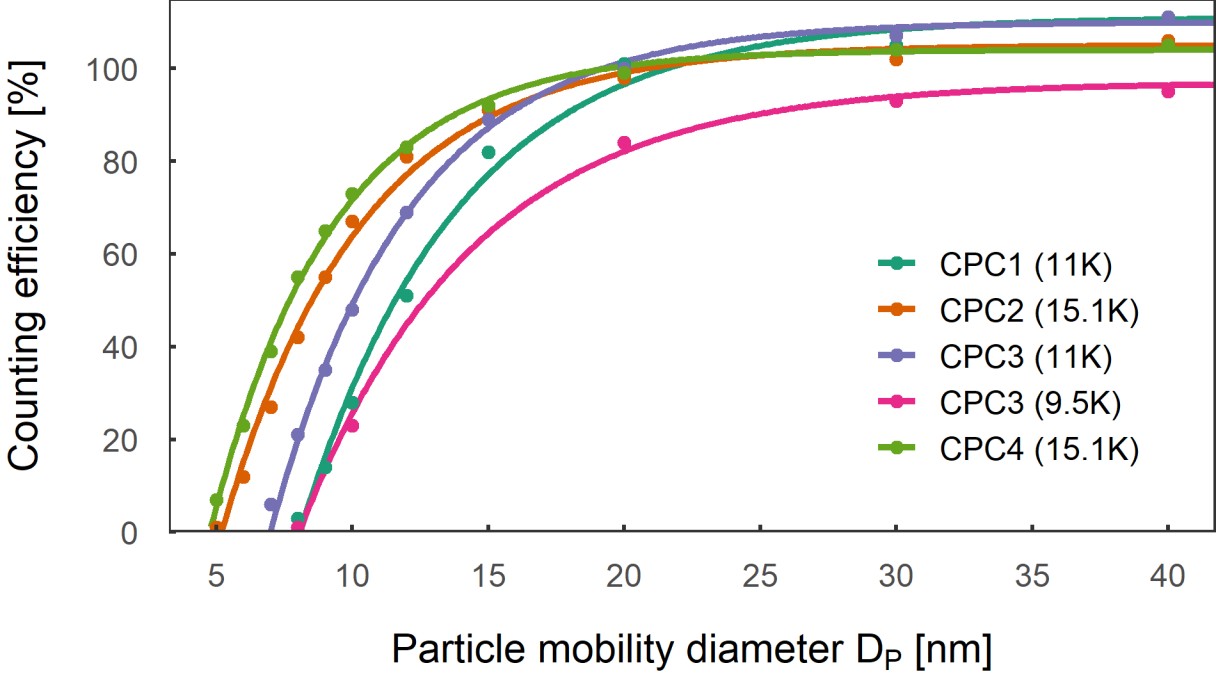

**Figure 2** Regression curves of counting efficiency of four modified CPC 3007 for different Peltier voltage settings with resulting temperature differences between saturator and condenser in brackets.

CPC1 on standard settings of $\Delta T$ = 11 K showed a relatively high detection limit of 12.1 nm compared to CPC3 with 10.5 nm and former findings by Hämeri et al. (2002). A possible explanation could be slightly modified instrument characteristics caused by installing the temperature sensors for the measurements of resulting $\Delta T$ with CPC1. For CPC2 and CPC4, a $D_{P50}$ of 7.9 nm and 8.8 nm were found at $\Delta T$ = 15.1 K. At $\Delta T$ = 9.5 K, CPC3 featured a $D_{P50}$ of 12.4 nm with a reduced $Eff_{max}$ compared to the calibration run with $\Delta T$ = 11 K, which probably results from a decreased degree of supersaturation. However, a $\Delta T$ of

9.5 K is suitable to achieve a more significant difference in detection limits, including the determined counting efficiency correction factor (Table 2). The final settings for CPC1 and CPC2 on CAMP1 are at $\Delta T$ of 11 K and 15.1 K resulting in lower detection limits of 12 nm ($N_{12}$) and 9 nm ($N_9$), respectively. For CPC3 and CPC4 on CAMP2, $\Delta T$ was set to 9.5 K and 15.1 K for detection limits of 12 nm ($N_{12}$) and 8 nm ($N_8$), respectively. A decrease in counting efficiency was found for the maximum setting of $\Delta T$ = 15.1 K at ambient temperature below 20°C. CAMP accounts for this temperature sensitivity through the

temperature-controlled heating system.

**Table 2** Calibration results of four CPC 3007 for different Peltier voltage settings. The correction factors account for offsets in sample flow rates from the customized flow system and reduced supersaturation in the case of CPC3 at 1.2 V. Uncertainties for the $D_{P50}$ are derived from a non-linear regression fit.

| CPC | Peltier setting [V] | $\Delta T$ [K] | $D_{P50}$ [nm] | $D_{P0}$ [nm] | $Eff_{max}$ [%] | Correction factor |
|-----|--------------------|-----------------|-----------------|----------------|------------------|-------------------|
| 1 | 1.4 | 11 | 12.1 ± 0.4 | 8.1 | 111 | 0.90 |
| 2 | 2 | 15.1 | 8.8 ± 0.2 | 5.2 | 105 | 0.95 |
| 3 | 1.4 | 11 | 10.5 ± 0.1 | 7.0 | 110 | 0.91 |
| 3 | 1.2 | 9.5 | 12.4 ± 0.5 | 8.0 | 97 | 1.03 |
| 4 | 2 | 15.1 | 7.9 ± 0.1 | 4.8 | 104 | 0.96 |

### 2.3 Optical Particle Size Spectrometer

The Portable Optical Particle Spectrometer (POPS, Handix Scientific Inc., USA) is a lightweight instrument for application on uncrewed airborne platforms (Boer et al., 2018; Creamean et al., 2020; Gao et al., 2016; Liu et al., 2021; Telg et al., 2017). Operating at 405 nm wavelength, the POPS measures the PNSD in an optical size range from 0.13 to 3 μm at 1 Hz resolution. In particular, the comparably low detection limit makes the POPS suitable for measurements in the Arctic. Sizing calibrations of two POPS were performed with test particles of polystyrene latex (PSL) spheres in 13 sizes ranging from 0.125 to 3 μm. In

parallel, the counting efficiencies against a reference CPC (model 3772, TSI Inc., USA) were analyzed for the sub-micron PSL sizes. Agglomerated PSL particles and residuals originating from the nebulizing process of PSL from a suspension were filtered

out with a differential mobility analyzer (DMA) (Heim et al., 2008) set to the mean $D_P$ of each PSL size. The calibration setup was arranged vertically to reduce losses of super-micrometer particles. On top of the setup, a nebulizer generated the test aerosol with particle-free air from a suspension. Subsequently, the particles were dehumidified by a silica dryer before entering

the DMA. A flow splitter with inbuilt core sampling per line distributed the dried PSL particles to the two POPS and the reference CPC. For data evaluation, lognormal distributions were fitted to the PNSD measured by the POPS with a size resolution of 200 bins using the particle diameter bin limits provided by the manufacturer. Sizing deviations of the optical particle diameter ($D_{PO}$) by the POPS from the mean PSL $D_P$ were defined by the mean mode of the lognormal fits and the sizing uncertainty from 1 standard deviation (SD) of the fit.

The results of the size calibration with PSL in Figure 3 a) show similar sizing deviations for the two examined POPS. Except for the PSL sizes between 0.5 and 1 µm, the sizing deviations were below 5 % for both units. In that particular particle size range, sizing deviations up to 27 % were found for both POPS. In addition, an increased sizing uncertainty of POPS1 ranging up to 23 % was seen in this size range in contrast to POPS2, showing a moderate SD of 8 %. Unit-to-unit variabilities caused POPS2 to produce a smaller SD yet with more significant sizing deviations than POPS1 with a larger SD but smaller sizing

deviations. The increased variations of both POPS for PSL sizes between 0.5 and 1 µm probably result from the geometric specifications of the instrument optics that cause Mie resonances to appear in the scattering amplitude (Gao et al., 2016). Concluding from the PSL calibrations, using the POPS in configurations of 32 size channels or more is not useful as the resulting bin sizes fall below the sizing uncertainties. In addition, the errors induced by Mie resonances are avoided by using 16 bins with one single size bin covering the size range from 0.6 to 1 µm.

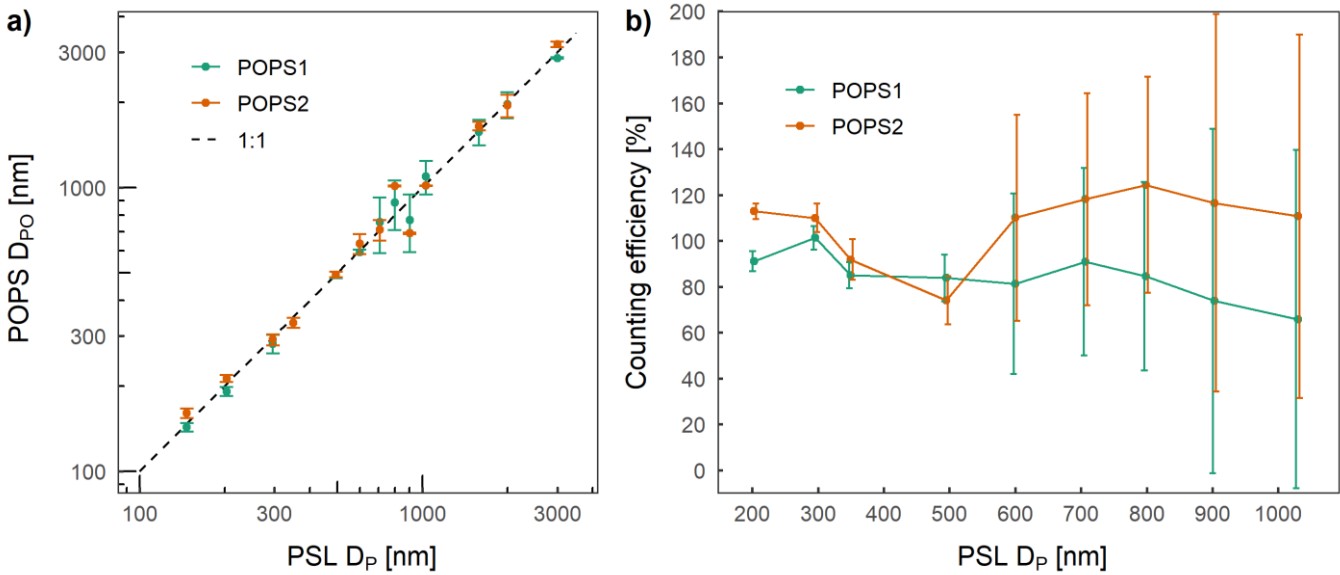

**Figure 3** a) Measured optical particle diameter ($D_{PO}$) by two POPS determined by lognormal fits on the measured particle number size distribution of mobility diameter ($D_P$) selected PSL particles. b) Counting efficiencies of the two POPS measured against a reference CPC during the PSL calibration.

The size-resolved counting efficiency of the two POPS was evaluated starting with determining the instrument noise level with

measurements of particle-free air. Integrated particle number concentrations across all size bins showed up to 10 cm$^{-3}$ for both POPS when no particles were present at a reference CPC. The analyzed PNSD exhibited increased concentrations in bins for particles below 150 nm contributing to more than 90 % of the overall noise. A possible explanation for this is the high sensitivity of the instrument light detector to Rayleigh scattering at air molecules or stray light from apertures (Gao et al., 2016; Mei et al., 2020). In Arctic environments with low particle number concentrations, the measurement uncertainties introduced

by noise in the lower bins are unacceptable. Consequently, bins below 150 nm were neglected for field measurements with the POPS on the two CAMPs. For the counting efficiency analysis, it was decided to focus on PSL sizes between 0.2 and 1 µm and evaluate the POPS particle number concentration as integral of the bins above 150 nm ($N_{>150}$). A detailed counting efficiency analysis for Ammonium-Sulfate particles with $D_P$ below 200 nm can be found in Mei et al. (2020). Figure 3 b)

illustrates the counting efficiency curves of the two examined POPS determined against a reference CPC with mobility size selected PSL particles. On average, POPS1 showed a lower counting efficiency of 83 % compared to POPS2 with 109 %, thus, revealing an evident inter-unit variability. Interestingly, both units exhibit a partial decrease in efficiency for PSL sizes between 0.3 and 0.6 µm, with POPS2 showing a more pronounced reduction of 35 % from the average at $D_P = 0.5$ µm. The increasing uncertainties with larger PSL diameters originate from the particle production from a suspension and the subsequent size selection with the DMA. However, the resulting average particle concentrations of 1 cm$^{-3}$ for 1 µm are comparable to ambient aerosol concentrations and the statistical relevance of the measurements is given by the comparison time of 1000 s. Other methods for producing higher concentrations (for instance from PSL powder) usually result in the creation of a polydisperse aerosol and were therefore not considered appropriate for the experiment.

## 2.4 Absorption photometer

The Single-channel Tricolor Absorption Photometer (STAP model 9406, Brechtel Manufacturing Inc., USA) is a filter-based instrument for the application on uncrewed airborne platforms (Bates et al., 2013; Pikridas et al., 2019; Telg et al., 2017). In principle, the STAP samples air through a filter on which aerosol particles deposit while the light intensity behind the filter ($I_s$) is measured at three wavelengths ($\lambda$) of 450, 525, and 624 nm. The deposited particles result in a light attenuation (ATN) that is derived in combination with a reference intensity behind a clean filter ($I_r$) as the logarithm of the ratio of the two light intensities denoted as the filter transmittance $\tau$ following Eq. (2):

$$\text{ATN} = -\ln\left(\frac{I_s(t)}{I_r(t)} \middle/ \frac{I_s(0)}{I_r(0)}\right) = -\ln(\tau) \tag{2}$$

At the instrument initialization ($t = 0$ s), the ATN is reset to zero by the initial clean filter transmittance. The change in attenuation $\Delta$ATN per time step $\Delta t$ is then related to the sample air column that passed through the filter during $\Delta t$ yielding the light attenuation coefficient for each wavelength ($\sigma_{\text{ATN}}(\lambda)$) following Eq. (3):

$$\sigma_{\text{ATN}}(\lambda) = \frac{A}{Q}\frac{\text{ATN}(t_2)-\text{ATN}(t_1)}{t_2-t_1} = \frac{A}{Q}\frac{\Delta\text{ATN}}{\Delta t}, \tag{3}$$

with the sample flow rate ($Q$) and the filter spot area ($A$). Corrections must be considered for filter loading and enhancement of $\sigma_{\text{ATN}}(\lambda)$ by multiple scattering and absorption of particles deposited on the filter. An empirically determined transmittance correction term $f(\tau)$ adapted from Bond et al. (1999) and Ogren (2010) accounts for quartz fiber filter loading and enhancement following Eq. (4):

$$f(\tau) = (1.0796\,\tau + 0.71)^{-1}. \tag{4}$$

In addition, Bond et al. (1999) and Ogren (2010) introduced two constants, $K_1 = 0.02 \pm 0.02$ and $K_2 = 1.22 \pm 0.2$, to correct for loading and enhancement by deposited scattering and absorbing particles, respectively. The particle light absorption coefficient ($\sigma_{\text{abs}}(\lambda)$) from the STAP measurements is then calculated following Eq. (5):

$$\sigma_{\text{abs}}(\lambda) = 0.85\frac{f(\tau)}{K_2}\frac{A}{Q}\frac{\Delta\text{ATN}}{\Delta t} - \frac{K_1}{K_2}\sigma_{\text{sca}}(\lambda), \tag{5}$$

with the particle light scattering coefficient ($\sigma_{\text{sca}}(\lambda)$) measured with a Nephelometer. An equivalent BC mass concentration ($m_{\text{eBC}}$) can be derived from $\sigma_{\text{abs}}(\lambda)$ with the wavelength-dependent mass absorption cross-section (MAC($\lambda$)) following Eq. (6):

$$m_{eBC} = \frac{\sigma_{abs}(\lambda)}{MAC(\lambda)}. \tag{6}$$

Two STAP units intended for the application onboard the two CAMPs were examined in the following section. Different filter materials can be used with the STAP, Pallflex filters (E70-2075W, Pall Cooperation, USA), and Azumi filters (371M, Azumi Filter Paper Co., LTD, Japan). Düsing et al. (2019) reported a sensitivity of the STAP equipped with Pallflex to quick relative humidity changes by water adsorption and evaporation and provided a correction function. The Azumi filters probably show the same sensitivity due to the similar composition of glass fibers. However, the sensitivity could be less pronounced for the Azumi without the hydrophilic cellulose backing material of the Pallflex. In a comparison by Ogren et al. (2017), Azumi filters were found to increase $\sigma_{\text{ATN}}(\lambda)$ by 25 % compared to Pallflex for measurements at ambient urban aerosol with the Continuous

Light Absorption Photometer (CLAP, NOAA, USA), which is the stationary version of the STAP. Since the Pallflex filters are no longer commercially available, we decided to use the Azumi filters with the STAP for this study to allow for comparison with future studies. A correction factor of $1.25^{-1}$ was applied for data evaluation to account for 25 % $\sigma_{ATN}(\lambda)$ enhancement by the filter material.

In the context of $m_{eBC}$ measurements in the Arctic atmosphere with an absorption photometer, the lower detection limit defined by the noise level of the instrument is vital. Bates et al. (2013) determined the STAP detection limit to 0.2 Mm$^{-1}$ at a 60 s average, which converts into $m_{eBC}$ of 15 ng m$^{-3}$ using an average MAC of 13.1 m$^2$ g$^{-1}$ at $\sigma_{abs}$(550 nm) obtained from long-term measurements at four Arctic sites (Ohata et al., 2021). Knowledge about averaging time-dependent noise levels is crucial for airborne measurements to find an appropriate compromise between high spatial coverage resulting from short averaging times and low detection limits at extended averaging periods. Therefore, a noise analysis was performed with one STAP during the CAMP developments. The instrument noise was defined as 1 SD of $\sigma_{abs}(\lambda)$ measured during a 20-hour sampling of particle-free air on a clean Azumi filter. The internal averaging interval ($\Delta t$) was set to 1 s, and the time series of $\sigma_{abs}(\lambda)$ was calculated according to Eq. (5). The SD was evaluated for varying $\Delta t$ from 1 to 180 s using arithmetic and centered moving averages.

The analysis results in Figure 4 show a distinct wavelength-dependency of the STAP noise level. The centered moving average resulted in a slightly lower SD than the arithmetic average, particularly for $\sigma_{abs}$(624 nm) at $\Delta t$ larger than 60 s. The STAP detection limit can be approximated by 9 Mm$^{-1}$ ($\Delta t$/s)$^{-1}$ for $\sigma_{abs}$(450 nm), 14.5 Mm$^{-1}$ ($\Delta t$/s)$^{-1}$ for $\sigma_{abs}$(525 nm), and 12.4 Mm$^{-1}$ ($\Delta t$/s)$^{-1}$ for $\sigma_{abs}$(624 nm) when using a centered moving average. In conclusion, the averaging time must be larger than 60 s to achieve a 0.2 Mm$^{-1}$ detection limit at all wavelengths. This requires individual data post-processing since the STAP firmware only allows for a maximum averaging time of 60 s. Occasionally varying ambient temperatures during the experiments indicated an increasing noise level with temperature. However, further investigations were unnecessary because of the controlled heating system inside CAMP. Other averaging methods that can improve temporal coverage by lower averaging times (Hagler et al., 2011) were not considered because of the capability of balloon-borne measurements to hover on constant altitudes, thus, enabling high averaging times when required.

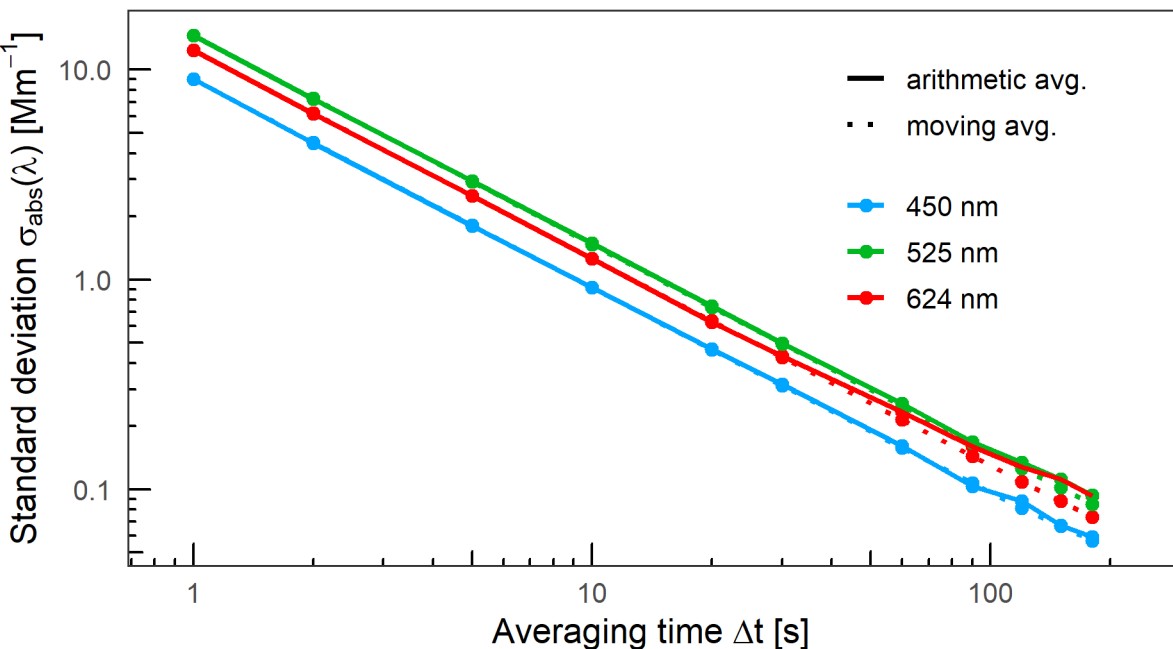

**Figure 4** Standard deviation of $\sigma_{abs}(\lambda)$ in dependency of the averaging time ($\Delta t$) measured by STAP during sampling of particle-free air analyzed with arithmetic and centered moving averages.

Having defined the lower detection limit of the STAP, a laboratory evaluation of the measurement performance at low $m_{eBC}$ was carried out. Two STAP with Azumi filters were compared against two stationary filter-based absorption photometers in a laboratory setup according to Müller et al. (2011), a Multi-Angle Absorption Photometer (MAAP 5012, Thermo Fisher

Scientific Inc., USA) and an Aethalometer (AE33, Magee Scientific, Slovenia). All instruments sampled ambient urban aerosol from a mixing chamber on three consecutive days over 16 hours. The daily sampling routine started with pure ambient air

progressively diluted with particle-free air up to 100 % and vice versa to simulate low $m_{eBC}$. For data evaluation, all instruments were averaged over 120 s. The $\sigma_{abs}(\lambda)$ measurements of the STAP units were calculated using Eq. (5) with an Azumi filter correction factor of $1.25^{-1}$. Because no Nephelometer data was available, $\sigma_{sca}(\lambda)$ was calculated from the ambient PNSD in the range from 10 nm to 800 nm measured by a mobility particle size spectrometer (MPSS, TROPOS, Germany) based on Mie-Theory. An urban aerosol refractive index of $1.51+0.01i$ was assumed (Alas et al., 2019). The disregarded scattering by

particles larger than 800 nm in the Mie-calculations can introduce uncertainties in $\sigma_{sca}(\lambda)$ up to 15 % (Virkkula et al., 2011), which is not considered a significant error in Eq. (5). Truncation was simulated for submicron particles according to Anderson and Ogren (1998) to get the $\sigma_{sca}(\lambda)$ from the Nephelometer in Eq. (5) with an absorption Ångström exponent (AAE) derived from the AE33 at $\lambda = 470$ and 660 nm by Ångström law following Eq. (7):

$$\sigma_{abs}(\lambda) = \sigma_{abs}(\lambda_0) \left(\frac{\lambda}{\lambda_0}\right)^{-AAE}, \tag{7}$$

To compare $\sigma_{abs}(\lambda)$ from the MAAP and AE33 measured at different wavelengths, the STAP $\sigma_{abs}(\lambda)$ were interpolated to $\lambda = 470$, 520, 637, and 660 nm with an AAE derived from the ambient mean $\sigma_{abs}(450 \text{ nm})$ and $\sigma_{abs}(624 \text{ nm})$ of each unit. $\sigma_{abs}(637 \text{ nm})$ was calculated from $m_{eBC}$ by MAAP with a MAC of 6.6 m² g⁻¹ according to Müller et al. (2011) following Eq. (8):

$$\sigma_{abs}(637nm) = 1.05 \, m_{eBC} \, \text{MAC.} \tag{8}$$

For the AE33, $\sigma_{abs}$ at 470, 520, and 660 nm were derived from $m_{eBC}(\lambda)$ by Eq. (6) with the $\lambda$-dependent MAC of 14.54, 13.14, and 10.35 m² g⁻¹ (Drinovec et al., 2015), respectively, and interpolated to $\sigma_{abs}(637 \text{ nm})$ by Eq. (7) from $\lambda = 470$ and 660 nm. Absorption measurements from the AE33 are systematically higher than the MAAP (Collaud Coen et al., 2010). Therefore, we derived a harmonization factor of $1.81^{-1}$ from a linear regression between the MAAP and the AE33 at $\sigma_{abs}(637 \text{ nm})$. The factor accounts for enhanced $\sigma_{abs}(\lambda)$ measurements by AE33 for comparisons with the STAP.

The results of the laboratory comparison in Figure 5 show an underestimation of 5% by STAP1 and a 1:1 agreement of STAP2 with the MAAP at $\sigma_{abs}(637 \text{ nm})$. At $\sigma_{abs}(470 \text{ nm})$ STAP1 showed 1 % and STAP2 4 % lower values; at $\sigma_{abs}(520 \text{ nm})$ STAP1 showed 10 % higher values and STAP2 a 1:1 agreement, and at $\sigma_{abs}(660 \text{ nm})$ STAP1 showed 7 % lower values and STAP2 a 1:1 agreement with the AE33. All linear regressions featured an $R² \geq 0.99$ and were forced through zero because no significant offset from zero was observed during measurements at particle-free air. The results confirm $\sigma_{abs}(\lambda)$ enhancement of 25 % by

Azumi filters with the STAP as Ogren et al. (2017) found for the CLAP and support the use of a correction factor of $1.25^{-1}$.

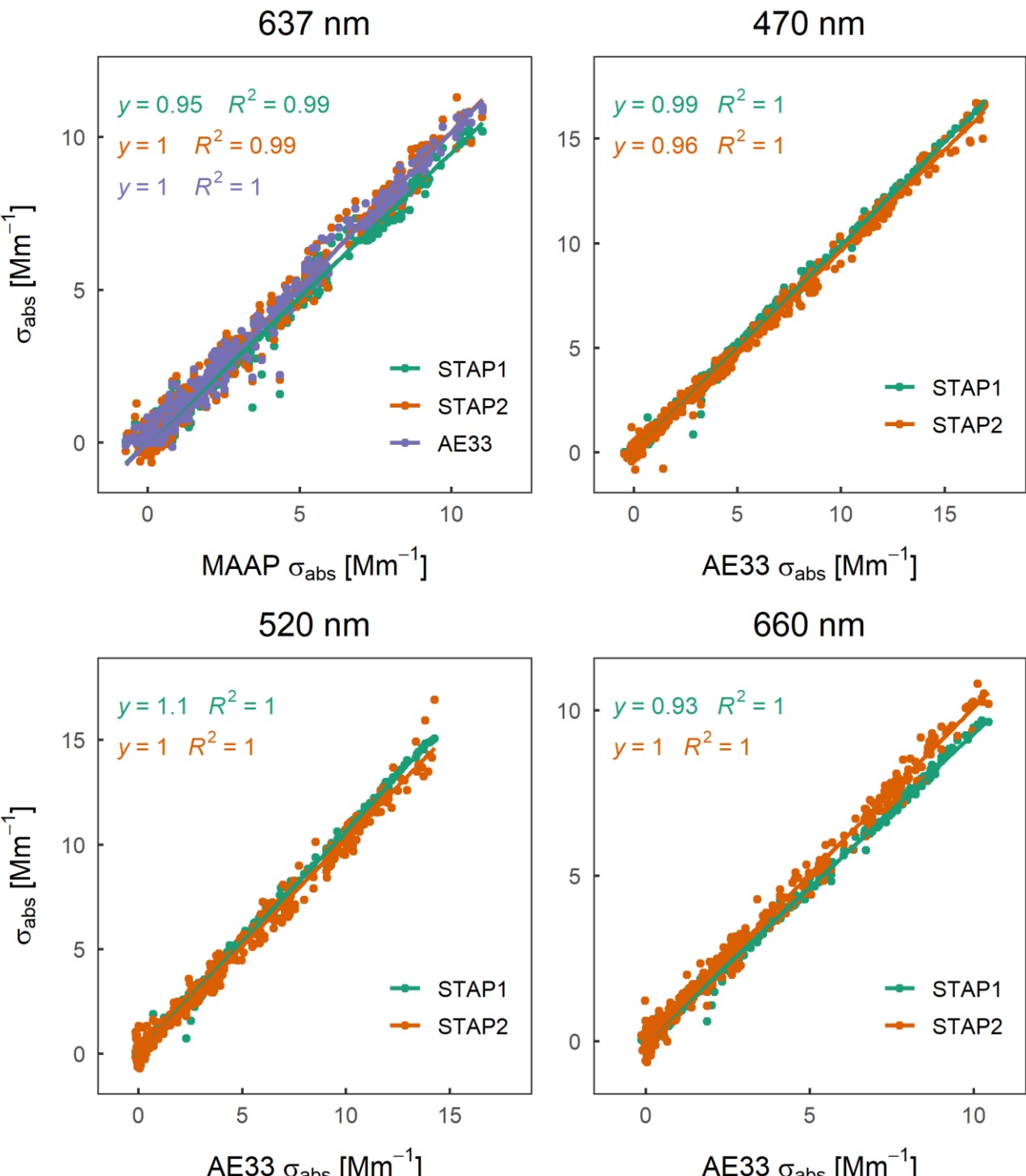

**Figure 5** STAP measurements of particle light absorption coefficient $\sigma_{abs}(\lambda)$ of ambient urban aerosol variably diluted with particle-free air compared with MAAP and AE33.

### 2.5 Aerosol sampling system

The stainless steel funnel inlet is vertically oriented to assure an omnidirectional sample air inflow for the CAMP system that is not pointed into the wind direction for instance by a wind vane. An opening angle of 30° increases sampling efficiency compared to a tube while a cover on top of the funnel protects against falling precipitation (Figure 1 and S1). Droplets or ice crystals inside clouds are not driven into the inlet at a prevalent inlet air velocity of 0.2 m s$^{-1}$ at the funnel opening. The interstitial inlet is not actively heated. However, the inlet's lower part outside of the platform is usually above ambient

temperatures because the lower end reaches into the heated platform.

The inlet aspiration efficiency depending on ambient wind speed ($u$) and particle diameter was calculated following Baron and Willeke (2001, hereinafter abbreviated as B&W) Eq.8-22 for a 90° sampling angle and a total inlet flow of 1.82 l min$^{-1}$ at an inlet diameter of 1/4" at the bottom of the funnel (Figure 6). An apparent inlet cut-off (efficiency below 50 %) that is below the POPS' upper detection limit of 3 µm results at ambient wind speeds above 2.5 m s$^{-1}$. With increasing wind speeds up to the 10 m s$^{-1}$ operational limit of many TBS systems, the cut-off diameter decreases to 0.5 µm. The theoretically determined inlet efficiency does not account for deviations from a 90° sampling angle resulting from an inclined CAMP system being attached to the tether during balloon deployments. Increased sampling efficiency due to the funnel geometry that is also not included in the calculations partially counterbalances these additional losses.

The custom silica-based diffusion dryer consists of a 170 mm long carbon fiber composite tube with a 40 mm outer diameter (Figure S1). 1/4" pipe connectors (Swagelok, USA) are threaded into the endcaps on both sides of the dryer. The sample flow runs through a straight stainless steel meshed 1/4" tube at the center of the dryer enabling humidity exchange to the silica beads (Merck KGaA, Germany). A 1/4" electrically conductive tube connects the inlet with the dryer. Downstream of the dryer is a 1/8" core sampling system integrated into a 1/4" T-connector to decrease diffusional losses (Fu et al., 2019). The bypass flow of the core sampling is used as sheath air for the POPS. All subsequent lines to the individual instruments are conductive 1/8" tubes separated from the main line with stainless steel T-connectors. The lines were kept as short as possible to reduce particle losses due to diffusion that result for each instrument line individually from different tube lengths and varying sample flows. The sampling efficiency per instrument line was calculated by accounting for impaction losses in 90° bents (B&W Eq.8-66 to 67), gravitational settling in horizontal lines (B&W Eq.8-51 to 53), diffusional losses (B&W Eq.8-60), and inlet losses (B&W Eq.8-22). Line specifications can be found in the technical sketch in Figure S1. Diffusional losses inside the dryer were determined by an estimated equivalent pipe length of 2.5 m and 0.15 m for 90° bents (Wiedensohler et al., 2012). Losses in the 1/8" T-connectors that separate individual sampling lines from the main line are calculated by B&W Eq.8-22. All calculations were performed for standard atmospheric conditions and a particle density of 1600 kg m$^{-3}$.

The results in Figure 6 show the highest sampling losses for the CPCs due to their low sample flow rates of 0.11 l min$^{-1}$ and relatively long sampling lines. The 1/8" T-connector that separates the POPS line from the main line represents a virtual impactor for particles larger than 3 µm because of the low sample flow ratio of 0.1. This is also the case for the separating connector to the CPCs with only a slightly higher sample flow ratio. However, these losses are neglectable concerning the upper detection limits of both instruments.

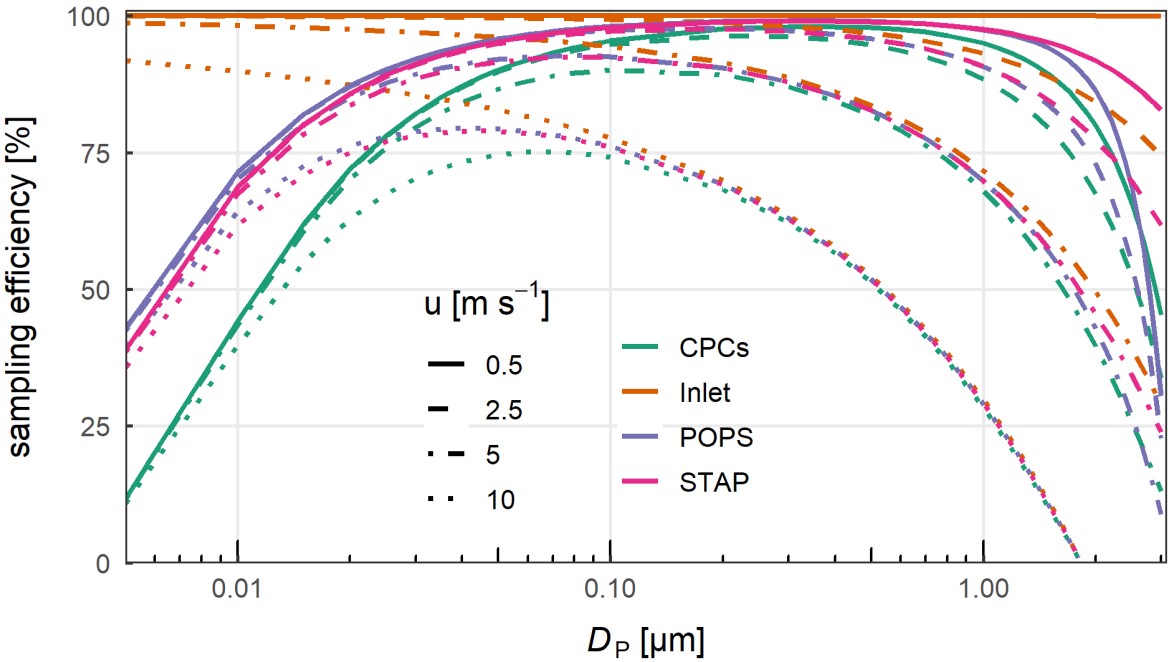

**Figure 6** Sampling efficiency of the funnel inlet alone and the sampling lines to the individual instruments including losses due to diffusion, gravitational settling, impaction, and at the inlet. Calculations are based on sampling line geometry and flow specifics of each instrument (Figure S1) as a function of particle diameter ($D_P$) and ambient wind speed (u).

## 3 First field deployment of CAMP

### 3.1 Measurement site and experiment

The first feasibility study with CAMP in the field was performed together with the BELUGA TBS at the TROPOS research station in Melpitz, Germany, in January and February 2019. CAMP was attached around 30 m below the balloon in combination with another sensor package 10 m above to measure standard meteorological parameters (*T*, *p*, RH). During profiling, the climb rates were typically below 2 m s$^{-1}$. Observations were made inside and outside of clouds up to maximum altitudes of 1.5 km. On five days of operation, 14 test flights were performed with CAMP in varying instrument configurations, with the data radio transmission not being operational yet. The complete instrument setup comprising CPC1 and CPC2, POPS1 in 16-bin configuration, and STAP2 was operated on two days with two flights each.

Particle number concentrations in the size range from 9 to 12 nm ($N_{9-12}$) and from 12 to 150 nm ($N_{12-150}$) were derived by the difference between the two CPCs, and CPC1 and the POPS, respectively. The barometric height ($h_b$) during balloon flights is calculated from the barometric pressure $p_b$ for data analysis following Eq.(9):

$$h_b = \frac{T_0}{L_0} \left(1 - \frac{p_b}{p_0}\right)^{\frac{L_0 R}{g}} \tag{9}$$

with the ground temperature $T_0$, the standard adiabatic lapse rate $L_0 = 6.5$ K km$^{-1}$, the ground pressure $p_0$, and the gas constant for dry air $R = 287$ J kg K$^{-1}$. All particle measurements were corrected to standard conditions of 273.15 K and 1013 hPa. Ground-based long-term measurements at the Melpitz site served as a reference for the performance evaluation. The observations covered meteorological parameters, the PNSD by an MPSS (TROPOS built) for mobility particle diameters from 5 to 800 nm, the PNSD by an aerodynamic particle sizer (APS 3321, TSI Inc.) for diameters from 0.5 to 10 µm, and $\sigma_{abs}$(637 nm) by a MAAP. The performance of CAMP's heating system and the silica dryer are evaluated on the coldest day of deployments in section 3.2. For an instrument evaluation against the ground station in section 3.3 and a case study of vertical observations with the complete CAMP system in section 3.4, the study focuses only on the balloon deployments on 15 February.

### 3.2 Heating system and dryer performance

The coldest conditions during the field deployments were seen on 21 January. At average ambient temperatures of -2.4°C throughout a balloon flight up to 1.2 km, the mean temperature inside CAMP was at 23.4°C with an SD of 1.2°C (Figure 7 a)). On the same flight, the ambient RH varied between 12 and 85 % while sample air RH downstream of the silica dryer inside CAMP was on average at 8.1 % with a SD of 2.5 % (Figure 7 b)). A low-level fog layer was also present during the balloon flight.

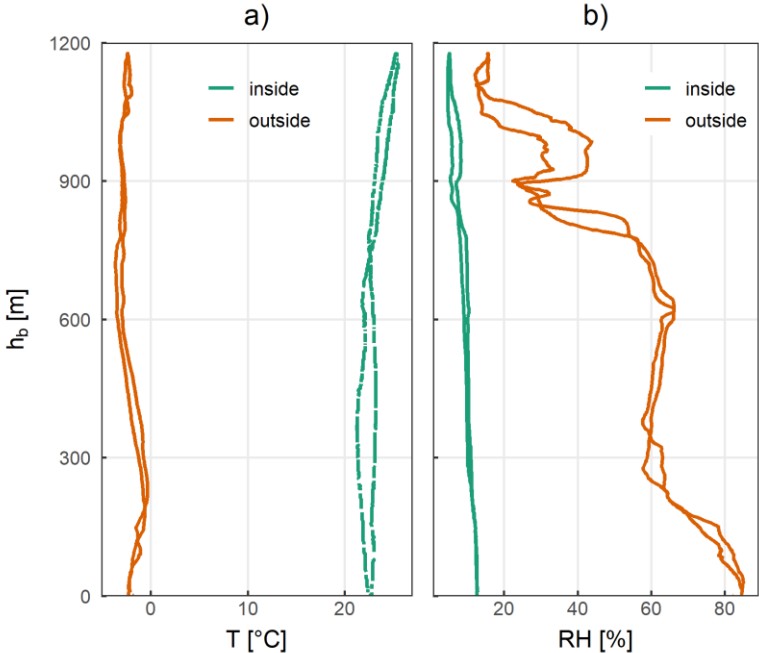

**Figure 7** Vertical profiles of a) temperature ($T$) and b) relative humidity (RH) inside CAMP downstream of the dryer measured on a balloon flight in Melpitz on 21 January 2019. Ambient measurements outside CAMP were taken from the external meteorological sensor package.

### 3.3 Comparison to the ground station

The measurements of the completely instrumented CAMP system were compared against the continuous observations from the Melpitz station on 15 February 2019. Three time series of CAMP observations were evaluated when the system was attached to the balloon at a constant height below 10 m before, in between, and after two flights. $N_{12}$, $N_9$, and $N_{>150}$ of CPC1, CPC2, and the POPS1 were compared with the integrated PNSD from MPSS for the respective size ranges at 20 min scanning times. CPC1, CPC2, and the POPS were corrected for the individual instrument counting efficiencies determined in the laboratory calibrations (sections 2.2 and 2.3). The three sampled time series were in-between MPSS scans. Therefore, the CAMP samples were averaged over the length of the time series (8, 6, and 15 min) while the MPSS was averaged over the result of the previous and past scans of each sample. $\sigma_{abs}$(624 nm) by the STAP was referenced to $\sigma_{abs}$(637 nm) by MAAP at 1 min averaging times. A 60 s centered moving average was applied to the STAP before averaging both instruments over the individual sample time.

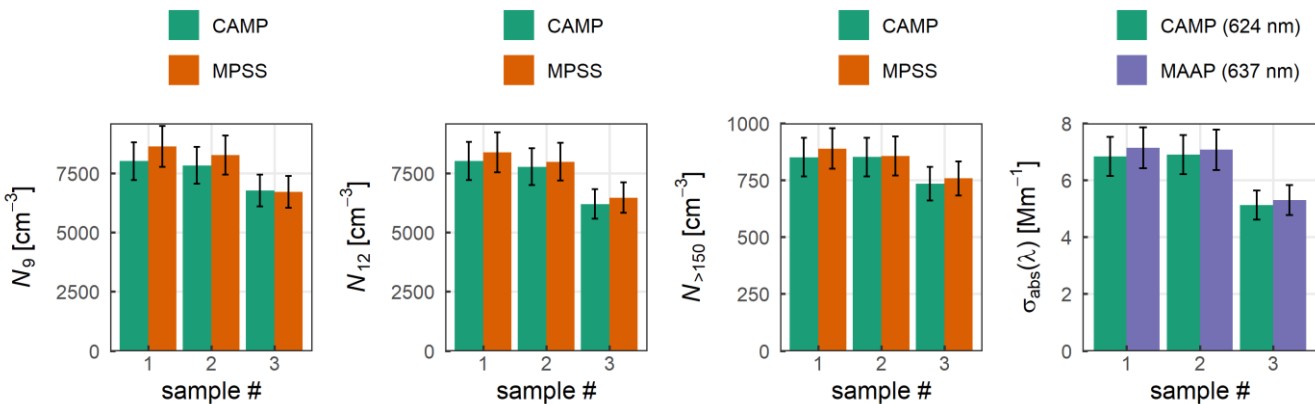

**Figure 8** Comparison of averaged particle number concentrations $N_9$, $N_{12}$, and $N_{>150}$ and particle light absorption $\sigma_{abs}$(624 nm) with integrated concentrations from MPSS and $\sigma_{abs}$(637 nm) from MAAP for three time series when CAMP was attached to the balloon on a constant height below 10 m on 15 February 2019.

The results in Figure 8 show an efficiency of CPC1 ($N_{12}$) between 95 and 97 %, of CPC2 ($N_9$) between 93 and 100 %, of the POPS ($N_{>150}$) between 95 and 99 %, and of the STAP ($\sigma_{abs}$(624 nm)) between 96 and 98 %. An average 96 % sampling efficiency at an average ambient wind speed of 3 m s⁻¹ (Figure S2) is within the range of the theoretical loss calculation from section 2.5 for the prevailing ambient aerosol (Figure S3). Still, the performed comparison is not fully representative for the sampling performance of the CAMP system in the field due to the short time series, the spatial distance the station inlet of 100 m and possible small-scale features around the platform or the station inlet. However, this study intends to present the feasibilities of the CAMP system in general and for future field deployments, an intercomparison to the individual ground station is mandatory and best performed closest to the station inlet.

A comparison of the optical PNSD detected by the POPS with the PNSD by MPSS and APS over a 20 min scanning interval is shown in Figure 9. The POPS was corrected for counting efficiency (see sec. 2.3), and the highest and lowest size bins were neglected because of inaccuracy. The aerodynamic particle diameter from APS was converted into mobility diameter concerning a shape factor of 1.1 (DeCarlo et al., 2004) and a particle density of 1.6 g cm⁻³ (Poulain et al., 2014). The optical PNSD by POPS based on PSL shows a significant underestimation of particles with diameters above 0.22 µm compared to the mobility PNSD by MPSS and APS. To some extent, the underestimation by the POPS possibly comes from the partial counting efficiency decrease of the instrument, which was determined in the laboratory calibration (see section 2.3).

Another reason for the underestimation is the difference between the optical diameter detected by the POPS based on the optical properties of PSL, which was used for calibration and ambient aerosol particles. To convert the optical PNSD to mobility PNSD, the optical properties of ambient particles have to be considered. Primarily, the complex refractive index ($\tilde{n}$) determines the optical properties of ambient aerosol when assuming a spherical shape of sub-micrometer particles (Alas et al., 2019). The theoretical response of the POPS to various refractive indexes was simulated based on Mie-theory with the geometric bin sizes provided by the manufacturer and the instrument optics specifications (Gao et al., 2016). A value of 1.5 was assumed for the real part of the refractive index, and for the imaginary part, values were varied between 0 and 0.02 $i$ for best fit with MPSS and APS, similar to Zieger et al. (2014).

The PNSD by POPS converted into mobility diameter with a refractive index of $1.5 \pm 0.02i$ in Figure 9 showed the best qualitative agreement with the other instruments for sub-micrometer particles. For larger particles, the correction resulted in an artificial overestimation of particle sizes and concentrations compared to the APS. However, the $\tilde{n}$-correction intends to highlight possibilities to match the optical PNSD of the POPS with the MPSS, for instance, to derive particle mass or volume. The simulated $\tilde{n}$ for best fit might not represent the actual ambient aerosol particle properties. A different $\tilde{n}$ would be required to represent better the optical properties of super-micrometer particles (Alas et al., 2019). In addition, non-spherical and irregularly shaped particles have to be considered when comparing coarse mode PNSD from an OPSS with an APS.

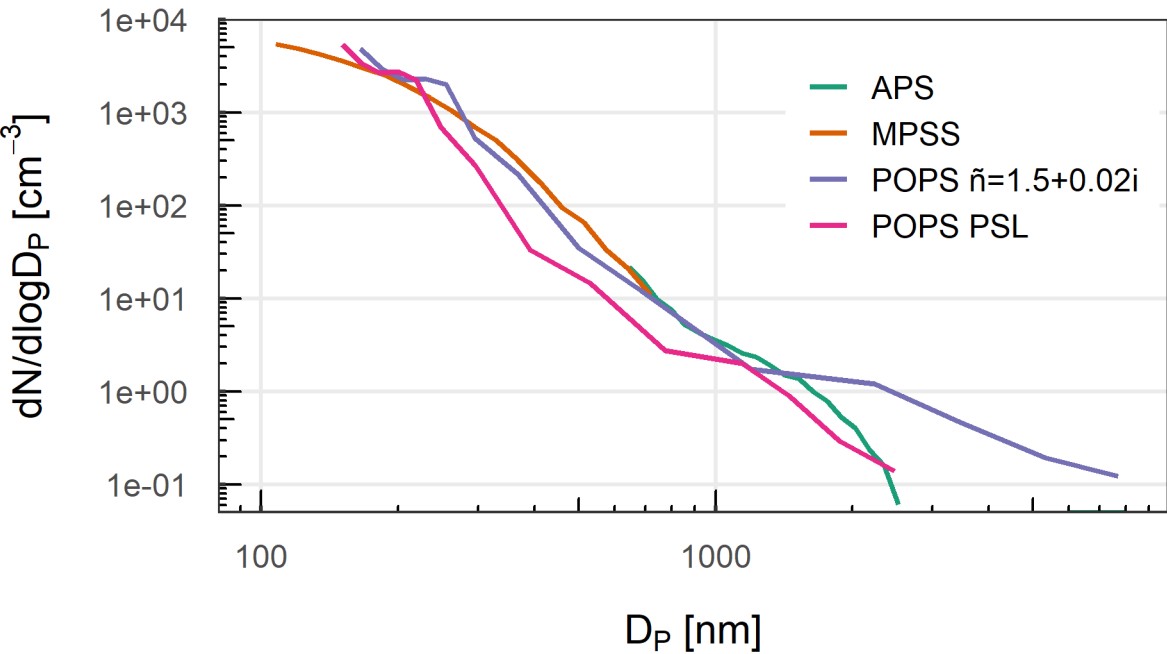

**Figure 9** Mean particle number size distribution (PNSD) of POPS on the ground compared with MPSS and APS over a 20 min scanning period in Melpitz on 15 February 2019. The PSL calibrated optical PNSD from the POPS was corrected for best fit with an assumed aerosol refractive index of $ñ=1.5+0.02i$.

### 3.4 Case study

Two balloon flights (i.e. four profiles) were performed in the late morning hours on a cloud-free day from 9:20 to 10:30 UTC and from 10:40 to 11:45 UTC on 15 February 2019. The ground-based meteorological observations in Figure S2 show a steadily increasing global radiation ($G$) up to 470 W m$^{-2}$ and $T$ from -2 up to 13°C while RH decreased from 96 to 50 % under low westerly winds ranging from 0.3 to 3.6 m s$^{-1}$.

The observed vertical profiles of potential temperature ($\theta$) and water vapor mixing ratio ($q$) derived from the balloon-borne meteorological sensor package are shown in Figure S3. The $\theta$ profiles depict an almost neutrally stratified ground layer of 150 to 200 m. The well-mixed ground layer slowly warmed up through convective heating. A temperature inversion on top of the mixed-layer gradually weakened from 0.02 to 0.01 K m$^{-1}$ while lifting. Above the inversion, a stably stratified layer up to 600 m showed a gradual shift towards neutral stratification with increased humidity fluctuations higher up.

The time series of integrated aerosol PNSD at the ground (Figure S4) showed decreasing trends in $N_{12-150}$ and $N_{>150}$ with minor variability between 9:00 and 12:00 UTC. $N_{9-12}$ appeared more variable with intermittent short-term rises in the meantime. In the afternoon, a distinct increase of $N_{9-12}$ occurred over three hours leading to doubling concentrations while $N_{12-150}$ and $N_{>150}$ continued to decrease further. The mean PNSD from 12:00 and 15:00 UTC showed a nucleation mode that is not present between 9:00 and 12:00 UTC (Figure S4). The PNSD time series does not indicate NPF at the ground as the source for the suddenly occurring nucleation mode particles.

Balloon-borne aerosol measurements by CAMP showed different particle distributions inside the well-mixed ground layer and the stable layer above on the two balloon flights (Figure 10 and Figure 11). Inside the stably stratified layer, particle layers between 300 and 450 m showed 5 times higher $N_{9-12}$ than the ground layer. Peak $N_{9-12}$ increased from 1000 to 3500 cm$^{-3}$ from the first to the third profile. In the meantime, the height of the layers with peak $N_{9-12}$ decreased from 430 to 310 m. Another shallow layer of increased $N_{9-12}$ was observed inside the temperature inversion on the second profile.

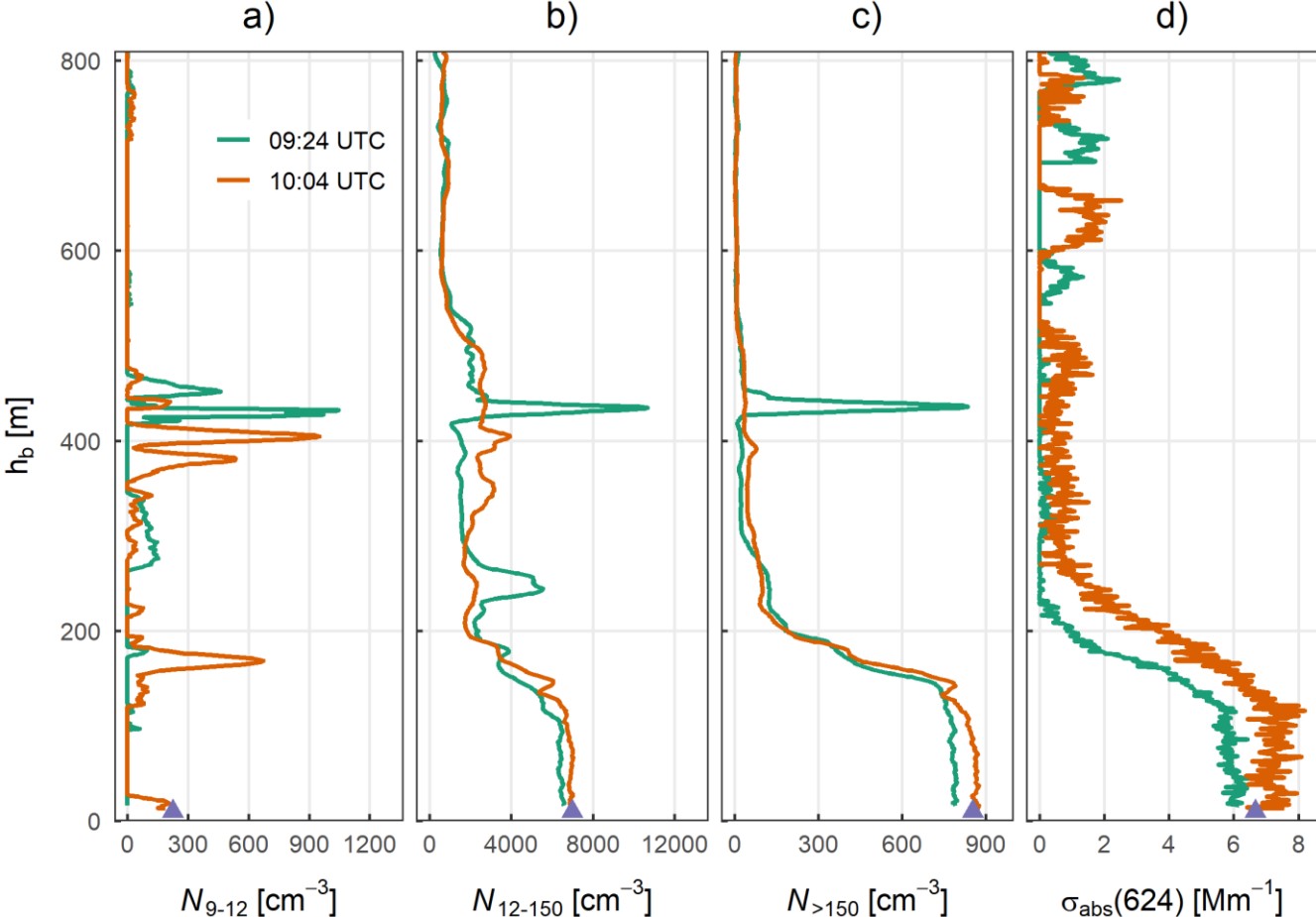

**Figure 10** Vertical profiles of particle number concentrations in size range from a) 9 to 12 nm ($N_{9-12}$), b) 12 to 150 nm ($N_{12-150}$), c) 0.15 to 3 µm ($N_{>150}$) displayed as 10 s centered moving averages, and d) of the particle light absorption coefficient at 624 nm wavelengths ($\sigma_{abs}$) displayed as 60 s centered moving average. The triangles represent the corresponding ground-based observations at 4 m inlet height of integrated $N_{9-12}$, $N_{12-150}$, and $N_{>150}$ from MPSS in a) to c) and of $\sigma_{abs}$(637 nm) from MAAP in d) at the start of the first profile. The displayed times represent the start of the ascent or descent profile during the first balloon flight in Melpitz on 15 February 2019.

$N_{12-150}$ and $N_{>150}$ were almost constant in the well-mixed ground layer. Distinct negative gradients in $N_{12-150}$ and $N_{>150}$ at the top of the ground layer marked the transition to the stably stratified layer with a generally lower Aitken and accumulation mode particle abundance. The gradients in $N_{12-150}$ and $N_{>150}$ gradually decreased from -54 to -44 cm$^{-3}$ m$^{-1}$ and from -7.0 to -4.4 cm$^{-3}$ m$^{-1}$, respectively. Shallow layers of increased $N_{12-150}$ and $N_{>150}$ were observed inside the stable layer between 430 to 450 m on the first and between 330 and 375 m on the last profile. Peak $N_{12-150}$ and $N_{>150}$ were up to 2 times higher inside these two layers than in the ground layer.

The $\sigma_{abs}$(624) profiles reflect the general trend of the $N_{12-150}$ distributions with less vertical resolution due to the applied 60 s moving average. In the well-mixed ground layer, $\sigma_{abs}$(624) were relatively constant except for profile three, featuring a decreasing trend with a local minimum at 150 m height. The second $\sigma_{abs}$(624) profile seemed biased by a measurement offset inside the ground layer since ground observations agreed well with the first profile, and no distinct increase in $N_{12-150}$ or $N_{>150}$ was observed. In addition, the $\sigma_{abs}$(624) measurements above 550 m appeared to be influenced by ambient humidity changes as there were no apparent variations in $N_{12-150}$ or $N_{>150}$ at altitudes of $\sigma_{abs}$(624) changes.

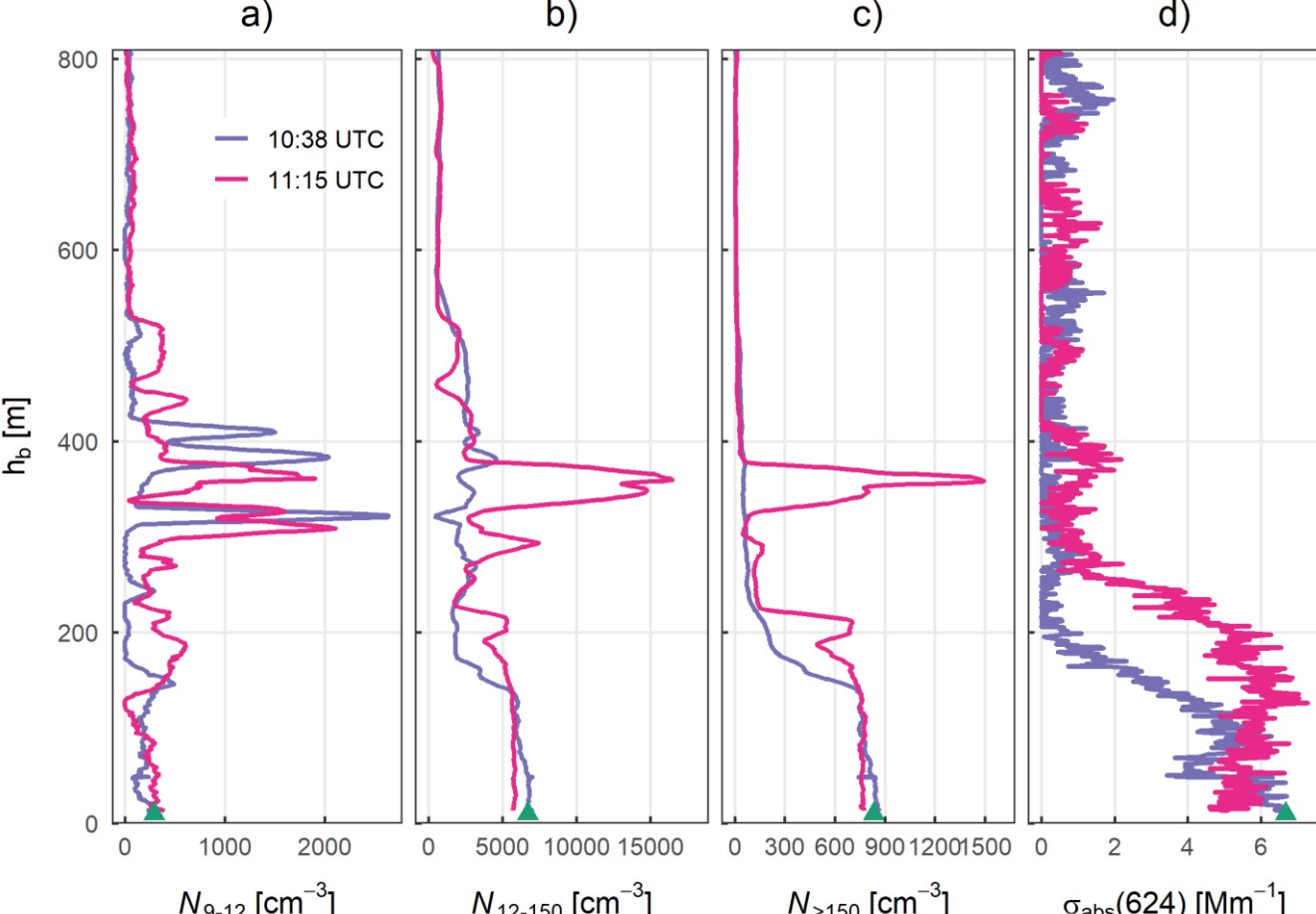

**Figure 11** Same as in **Figure 10** observed during the second balloon flight in Melpitz on 15 February 2019.

From the balloon-borne and ground observations, we conclude that the layers with increased $N_{9-12}$ above the well-mixed ground layer originated from new particle formation (NPF) and were mixed down after the balloon flight. Measured peak $N_{9-12}$ that corresponds to NPF events, the weakening temperature inversion, gradually decreasing gradients in $N_{12-150}$ and $N_{>150}$, and the increase in $N_{9-12}$ on the ground in the afternoon support this hypothesis. The occasionally appearing plumes of increased $N_{12-150}$ and $N_{>150}$ close to the layers of increased $N_{9-12}$ did not necessarily prevent NPF in the stably stratified layer, as shown by airborne observations at a different site by Wehner et al. (2010). In particular, the appearance of increased nucleation mode particle concentrations originating from NPF on top of a well-mixed ground layer, as seen on the second profile, was previously reported from airborne observations at Melpitz in summer (Platis et al., 2016; Siebert et al., 2004). Both studies showed that turbulence and the thermodynamic conditions inside an inversion layer could create favorable conditions for NPF. Another airborne study at Melpitz by Stratmann et al. (2003) observed NPF inside a residual layer and subsequent downward mixing of nucleation mode particles after the inversion breakup. Evaluations of long-term airborne observations above Hyytiälä (Finland) by Lampilahti et al. (2021) showed that NPF is likely to occur above mixed-ground layers and emphasized the importance of downward mixing after inversion breakup. The presented case study highlights possible NPF processes at higher altitudes and their implications for particle abundance on the ground at continental sites in winter.

## 4    Summary and Outlook

This study presented the newly developed CAMP for tethered balloon-borne aerosol particle observations. CAMP is designed to reliably provide observations of particle microphysical properties in cold and cloudy ABL like in the Arctic. The four instruments onboard CAMP are suitable to assess vertical distributions of Nucleation, Aitken, and Accumulation mode particle concentration, Accumulation mode PNSD, and particle light absorption. These observations enable process studies of ABL dynamics interacting with particle abundance aloft and on the ground, NPF processes at higher altitudes, and atmospheric

distribution of absorbing particles. The set of mobile devices was calibrated and characterized in laboratory studies to provide reliable in situ measurements. Commercially available handheld CPCs were modified to achieve different lower detection limits. Nucleation mode particles originating from NPF can be identified by the difference in particle number concentrations of the two CPCS. Improvements in the CPC flow systems resulted in a reduction of measurement uncertainty from 20 to 10 %. Two POPS units were calibrated with size-selected PSL particles and the results showed sizing uncertainties below 10 % when operating the instrument with 16 size bins. The mean counting efficiencies of 83 % and 109 % of the POPS were determined in the submicron particle range parallel to the calibration. For the STAP, the relation between averaging time and measurement noise was determined to enable the optimization of the temporal/spatial resolution with the required detection limit during data post-processing. A laboratory comparison of two STAP units with a MAAP and an AE33 at low eBC mass concentrations showed average deviations below 10 %.

CAMP was first tested in a field campaign with the BELUGA TBS at the TROPOS research station in Melpitz, Germany, in January and February 2019. The platform was operated in different instrument configurations on 14 balloon flights up to 1.5 km, under cloudy and clear-sky conditions, and at ambient temperatures between -8 and 15°C. The performance of the heating system and the dryer was evaluated on the coldest day during the campaign. A comparison of the fully instrumented CAMP system as presented in this study with the nearby Melpitz station was performed for one day of the field campaign. CAMP measurements of particle number concentrations by CPCs and POPS compared to an MPSS over 20 min averaging periods were within 10 % uncertainty. Observed particle light absorption coefficients by the STAP were within 10 % uncertainty of a MAAP for 1 min resolution. The POPS optical PNSD based on PSL was compared to MPSS and APS with an additional conversion to mobility PNSD by a refractive index correction for the POPS to find a best fit the other instruments. A detailed case study of CAMP observations from two balloon flights and their relations to ground-based measurements on 15 February 2019 was presented. The case study highlights the observational capabilities of the system and the high relevance of balloon-borne measurements for aerosol process studies in the lower atmosphere by establishing connections between ABL dynamics and ground based observations. The results of the laboratory instrument characterizations and the first field observations demonstrate CAMP's abilities to provide reliable aerosol particle observations under challenging environmental conditions. After the first field application, the CAMP capabilities were proven during measurements with BELUGA on the international Multidisciplinary drifting Observatory for the Study of Arctic Climate (MOSAiC) expedition in the summer of 2020 (Shupe et al., 2022). Ongoing analysis of the data collected on MOSAiC will be used for more detailed performance analysis of the CAMP system and providing new insight into aerosol distributions in the complex structured Arctic ABL. Further CAMP deployments were made in combination with a balloon-borne ice nucleating particle sampler at the research station Ny-Ålesund, Svalbard within the AC³ project (http://www.ac3-tr.de/).

## 5    Author Contribution

CP designed the CAMP platform with the support of BW and SD. CP, BW, TM, and SD designed the laboratory experiments and CP carried them out with the support of SD. HS, and BW organized the BELUGA field deployments and CP, ML, HS, BW, and JV carried them out. CP evaluated the data from the laboratory experiments and field observations with the support of SD. JV designed the meteorological sensor package for the balloon observations and processed the data. CP prepared the manuscript with contributions and revision from all co-authors.

## 6    Acknowledgement

We thank Kay Weinhold, Sascha Pfeiffer, and Maik Merkel for their technical support in the laboratory experiments. Special thanks go to Ralf Kaethner and the TROPOS workshops for technical and manufacturing support of CAMP. The authors thank Thomas Conrath for the technical support in the field with BELUGA during the Melpitz campaign. We highly appreciate the

constructive feedbacks of the two anonymous referees for improvements of the manuscript. We acknowledge the support by Deutsche Forschungsgemeinschaft (DFG, German Research Foundation) – Project-ID 268020496 – TRR 172.

## 7    Competing interests

The authors declare that they have no conflict of interest.

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
