# Peer review of "CAMP: an instrumented platform for balloon-borne aerosol particle studies in the lower atmosphere"

_Atmospheric Measurement Techniques, 2022_

## Author Response (AR1)

**Feedback comment on amt-2022-175**

**Referee #1**

**General comment**

Pilz et al. present in their manuscript a newly developed platform to measure aerosol properties on tethered balloon system. The platform, called CAMP, is especially designed to operate in environmentally challenging areas such as the Arctic. Vertical profiles of aerosol properties are indeed important parameters needed for all kinds of studies and model improvements and are indeed generally under-sampled (especially in the Arctic). The work therefore presents an important contribution to the experimental atmospheric science community, it is well written, and therefore, the manuscript is suitable to be published in AMT. However, some important technical details/comparisons are missing and need to be added before final acceptance. The manuscript also lacks a detailed error analysis of the entire setup, which is a prerequisite for publication in ATM. In addition, a few further (minor) clarifications and suggestions for improvements are listed below. Overall, I recommend major revisions.

**Detailed comments**

 As a technical paper, a detailed drawing of the CAMP system should be added which should include details on sizes/scales, tubing, tubing length, flow rates, etc. It would also be helpful to have a summarizing technical table of the different sensors with sampling rate, sensitivity, flow rates, etc.

 $\rightarrow$  a detailed sketch of the system was added to the supplementary. The sketch is considered too complex for the broader audience and therefore not adequate for the main article. Also, the sketch can be published in its original size in the supplementary

 $\rightarrow$  a table was added to section 2.1 as suggested

 Please add a few more important details on the inlet. How does it perform at high wind speeds? Is there an (apparent) size cut? Is it heated? How does it perform under cloudy conditions?

 $\rightarrow$  an additional section 2.5 "Aerosol Sampling System was added to describe details of the inlet, the dryer (requested by Referee #2), and the complete sampling system.

 The performance of the different sensors was evaluated individually but not with the sensors installed inside the CAMP system (if I understood it correctly). The particle losses could be much different. It is therefore warranted to add a detailed loss analysis (theoretically or experimentally) of the entire CAMP system (from the inlet to the sensor).

 $\rightarrow$  the complete system with all sensors installed inside the CAMP system is evaluated in section 3.3. The section was revised according to the additional comments on that section

 $\rightarrow$  An inlet and sampling system loss calculation is part of the new section 2.5. The losses are calculated for varying particle diameters and ambient wind speeds.

- Line 10: I would replace "to capture" with "to probe"
   → changed as suggested
- Line 13: Maybe add "of aerosols" behind "measurements"
   → considered as redundant since the previous sentence already states "observations
   of aerosol particle microphysical properties"
- Line 20-24: Since this is a technical paper, the last part of the abstract could be shortened by just stating that "first example profiles will be discussed to elucidate the performance of the system"
   → changed as suggested
- Line 39: Add "observations" behind "aircraft".
   → changed as suggested

- Some of the acronyms are not properly explained/introduced. For example, BELUGA or TROPOS
  - $\rightarrow$  acronyms introduced on the first appearance
- Throughout the text: the country of the manufacturer is usually added next to the company name and instrument model
- ightarrow all manufacturer info updated and country information added
- How was the height determined? Is there a GPS sensor or pressure installed?

   → height is determined by barometric pressure due to the higher accuracy compared to GPS altitude. Eq. 9 for barometric height calculation was added to section 3.1 and the y-axis caption of plots in section 3.4 was updated
- Figure 2: Is this a volume equivalent diameter? Could the difference in CPC3 and CPC4 also originate from different diffusional losses due to different tubing/flows? Please add information on the used tubing.

→ it is the mobility diameter selected by a DMA. The test particles are generated by a silver oven with a subsequent sintering process. The particles are almost spherical in shape. The x-axis description and the text are updated to mobility diameter, and a reference for the particle morphology of the calibration setup was added (Tuch et al. 2016) Line: 124.

The tubing length in the calibration setup and the internal tubing of the two CPCs inside CAMP are identical as well as the flow rates. Therefore, the determined counting efficiency curves with the resulting differences in detection limits are not considered to originate from differing diffusional losses in the tubing. However, the slightly different counting efficiencies of each instrument determined within the calibration originate from the software of each instrument. Therefore, both instruments are corrected to 100 % by the correction factors listed in Table 2.

- Sect 2.2: At what wavelength does the POPS operate?
   → at 405 nm wavelength, additional sentence added Line 155
- Figure 3 and 4 could be combined to a 2-panel figure since they are both related to the POPS to save some space.
   → figure updated as suggested
- Figure 4: Please add error bars.
   → figure updated as suggested
- Line 199: Where do these constants come from? Provided by the manufacturer or found in the literature?

 $\rightarrow$  they are part of the correction schemes by Bond et al. (1999) and Ogren (2010). Line 219 updated with literature info

 Line 215 (and later in the text): What is the reasoning behind writing 1.25-1 and not 0.8?

 $\rightarrow$  it intends to enable an easier connection to the described 25% enhancement by the filter material

• Line 248: The sentence on the 10% uncertainty due to the missing coarse mode needs a reference.

 $\rightarrow$  reference added as suggested, Line 267 updated

Figure 6 and line 266: What is the reasoning of forcing the linear regression through 0? You could miss constant off-set between the different sensors.
 → Zero checks for each individual instrument didn't show a significant offset from

→ Zero checks for each individual instrument didn't show a significant offset from zero. The offsets derived from regressions not forced through zero (plot below) are not considered significant because they are below the detection limit of the instruments. The experimental setup with varying ambient aerosol is not considered appropriate to determine offsets at these low concentrations. A BC generator as a constant aerosol source, at extended measurement times to avoid nonlinearities in filter correction schemes, and at absorption values below 0.5 Mm-1 would be required to determine a potential offset at about 0.1 Mm-1. This is far beyond the scope of the paper. The comparison intends to fill a gap in the literature for a general validation of the STAP's performance at very low concentrations compared with the most common reference absorption photometers in ground-based observatories. → Line 286 updated

What kind of R-value do you show? What is R\_adj?

 $\rightarrow$  Adjusted R2 is a corrected goodness-of-fit (model accuracy) measure for linear models. It identifies the percentage of variance in the target field that is explained by the input or inputs.

 $R^2$  tends to optimistically estimate the fit of the linear regression. It always increases as the number of effects are included in the model. Adjusted  $R^2$  attempts to correct for this overestimation. Adjusted  $R^2$  might decrease if a specific effect does not improve the model.

Adjusted R squared is calculated by dividing the residual mean square error by the total mean square error (which is the sample variance of the target field). The result is then subtracted from 1.

Adjusted  $R^2$  is always less than or equal to  $R^2$ . A value of 1 indicates a model that perfectly predicts values in the target field. A value that is less than or equal to 0 indicates a model that has no predictive value. In the real world, adjusted  $R^2$  lies between these values.

Source:

https://www.ibm.com/docs/en/cognos-analytics/11.1.0?topic=terms-adjusted-r-squared

For models with only one dependent,  $R^2$  and adjusted  $R^2$  do not vary much, which is the case for the here presented correlations. For clarity, the plot was updated with  $R^2$ .

 Line 273: "application" -> "deployment" → updated as suggested  Section 3 is a bit confusing and needs some slight refurnishing and a few clarifications. First, you mention that the system was tested in January and February 2019 but then you eventually only show one day of measurements. You could (a) show the entire comparison, e.g. how the instruments compared to the ground-based observation (as scatterplots) or how the CAMP system behaved under really cold conditions (which is not really shown but claimed) or (b) you reduce this section to just the one example day.

→ The first field deployment was a feasibility study with different configurations of the CAMP system. The complete system was only operated on two days with minor errors in data acquisition on one of the days. Therefore, the evaluation of the complete system and the case study focus on one day only, which is clarified in section 3.1. A new section 3.2 "Heating System and Dryer performance" was added to show the performance of the heating system on the coldest day of the test campaign.

- Line 290: What was the mean and STD of RH during the flights?
   → mean RH = 5 %, STD RH = 2.2 %, it is more detailed described in new section 3.2
- Line 295-299: As mentioned above, this finding is not really shown. How did you determine the 25%? Was it constant with time? Was it observed in all the instruments? What are the reasons?
   → this section was completely reworked. A failure was identified in the comparison

data during the review process. The instruments showed only 5% losses compared to the ground station. More details of the comparison, a figure with the results and a more detailed discussion was added to section 3.3.

Figure 7 is also only for the 15th of February. As such it should be moved with its discussion to Section 3.3

ightarrow it was moved to the same section as the comparisons to the ground station

- Figure 8, 9, 10 and 11 are not really CAMP-related and could be moved to the SI.
   → Figure 8, 9, 10, 11 are moved to SI as suggested
- Figure 10 and 11 could also be combined
   → combined as suggested to Figure S4
- Figure 12 and 13 are important as they show the first successfully recorded profiles. A few suggestions for improvements:

You could add the corresponding ground measurements to the figures as well (e.g. mean +/- std at height 0).

 $\rightarrow$  the corresponding ground measurements at start of the flights are displayed as asterisk and described in the figure caption. The plot was updated with a different point shape and color to highlight the station observations. Error bars for standard deviation are not visible due to the range of the x-axis scale.

Maybe it is better to replace the label N\_150 by N\_>150 (since everything larger than 150nm was measured).

 $\rightarrow$  updated as suggested

To demonstrate that the CAMP system is also capable to measure under harsh conditions (one of the main features), the temperature and RH-profiles should be shown as well (in the main text or at least in the SI).  $\rightarrow$  added to section 3.2

- Line 396: It is slightly inconsistent that you talk about eBC but you mainly show the particle light absorption coefficient in your profiles.
   → line rephrased to absorbing particles
- Paragraph starting with line 408: These claims are not really shown in your result section. Please revise.
   A revised to make clear that measurement performance was only evaluated for on

 $\rightarrow$  revised to make clear that measurement performance was only evaluated for one single day of field deployments

- I would suggest to rename Sect. 4 to "Summary and outlook" → renamed as suggested
- The last two paragraphs of the conclusion section could be shorted and combined. For example, by stating that first successful profiles were recorded (even with some interesting science in it!) and what the future will bring (analysis of your profiles

recorded at Melpitz or in the Arctic). Please keep in mind that AMT is a technical journal. What will further technical improvements could you think off? What would be the next steps in terms of instrument development?  $\rightarrow$  paragraph shortened as suggested

Many thanks for the very constructive and helpful feedback! It improved the manuscript a lot.

**Referee #2**

General comments:

The authors present an airborne measurement platform for aerosol measurements, the cubic aerosol measurement platform (CAMP). It has to be pointed out that the authors present solely the payload with mobile instrumentation inside and not the whole tethered balloon system (TBS), which I was expecting from the title. Within the manuscript authors put a great emphasis on laboratory tests of all mobile instruments. The manuscript reads well and contains a lot of valuable information, however since this journal is of technical matter, I would ask authors to provide more details in some parts.

→ The title was slightly changed to make clear that only a TBS payload is subject of the paper

**Specific comments:**

Could you describe in more detail the cube itself? Construction material and weight, insulation material choice, the inlet orientation, why vertical? Inlet cutoff, total flow, lenght, diameter and material, and similarly for the dryer.

- → More technical details were added to section 2.1 and a detailed technical sketch was added to the supplementary
- ➔ More information about the inlet and dryer including a loss calculation were added in a new section 2.5

The intercomparison measurements between CAMP and Melpitz are missing some detailed descriptions, it is not clear to me how it was done. Was the CAMP just sitting next to the station inlet, or was it attached to TBS and the CAMP was airborne at certain height? For airborne measurements the influence of the TBS itself might generate a bias, small turbulences, orientation of the inlet, iso-kinetic, iso-axial sampling and etc. might play a role, please discuss it. If the first case, then the ground-based sampling next to station inlet is a good start for intercomparison, but not a good reference for further measurements, e.g. comparison to a tower does a better job.

- → The comparison was done for CAMP being airborne on a constant height below 10 m on the balloon tether
- ➔ More details on the comparison and a new figure with the results were added to section 3.3

Figures 12 and 13, what is the height of Melpitz station inlet at which you make the comparison to it.

→ At 4m above ground, info added to figure description

---

## Author Response (AR2)

Author's response to

Associate Editor decision: Publish subject to technical corrections
by Jessie Creamean

**Comments to the author**:
The revision was sufficient to proceed with publication, subject to a few minor corrections that should be addressed:

• Please add a reference statement for Figure S1 in the caption of Figure 1.

→ added as suggested
• Lines 324-325: Why is the sampling efficiency necessarily increased through the funnel geometry? Couldn't the geometry also increase losses (e.g., through impaction)?

Compared to a simple pipe inlet, the funnel geometry enables more larger particles with higher inertia to be sampled as illustrated in the following sketch:

[Figure]

→ Larger particle would impact into pipe wall due
   to inertia without the funnel

The only possible scenario in which particle impaction would be increased is when the particle's initial flow direction is from below. In this case, the change in flight direction of 180° would most likely also lead to a loss of large particles that cannot follow the streamline of the sample air due to inertia with a simple pipe inlet.

• Line 333: Maybe add within this sentence "…due to diffusion..." behind "losses".

→ added as suggested
• Lines 191-192 and 350: Looks like there was an issue with the citation software not linking to the correct reference.

→ issue solved
• Line 356: Are these units supposed to be just % and not °%? If the latter, can the authors define that better?

→ that was a miss type for the "half-space" in MS word between number and unit, it is just %

→ miss type corrected

• Figure 9: Why was the POPS not converted from optical to mobility diameter to enable a direct comparison with the converted APS and MPSS? Please add a sentence or two providing more information on this, or how it might cause some discrepancies when comparing optical to mobility diameters.

→ the POPS was converted to mobility diameter by the refractive index correction that is described in the whole paragraph. For clarity, the paragraph was slightly modified to make clear, that the optical PNSD based on PSL requires a refractive index correction to be converted into mobility PNSD and compared to MPSS and APS.

• Line 467: eBC was not shown, perhaps change "eBC" to "absorption".

→ changed as suggested

Figure 1 was extended with a simplified sketch of the aerosol sampling system and instruments to enable an easy and brief overview of the system for the readers that are not interested in the detailed technical drawing in the supplementary.